# Novel Eel Skin Fibroblast Cell Line: Bridging Adherent and Suspension Growth for Aquatic Applications Including Virus Susceptibility

**DOI:** 10.3390/biology13121068

**Published:** 2024-12-20

**Authors:** Zaiyu Zheng, Bin Chen, Xiaodong Liu, Rui Guo, Hongshu Chi, Xiuxia Chen, Ying Pan, Hui Gong

**Affiliations:** 1Biotechnology Institute, Fujian Academy of Agricultural Sciences, Fuzhou 350003, China; cammy_zh@hotmail.com (Z.Z.); chenbinbin0721@163.com (B.C.); xdliu777@163.com (X.L.); chihongshu@faas.cn (H.C.); xiuxiachen@163.com (X.C.); py19860220@sina.com (Y.P.); 2Fuzhou Ocean and Fisheries Technology Center, Fuzhou 350007, China

**Keywords:** *Anguilla anguilla*, suspension growth, reversible adherent–suspension culture system, Anguillid herpesvirus, red-spotted grouper nervous necrosis virus

## Abstract

The adherent and suspension cell culture methods have their own advantages. Adherent culture involves both straightforward steps and inexpensive consumables, and the even spread of cells on the substrate facilitates clear observation of the cell morphology, growth status, and cytopathic effects using a microscope. By comparison, the ability to form a high-density cell population and the ease of scaling up make suspension culture valuable for industrial production, such as vaccine development, bio-pharmaceutical manufacture, medicine screening, and food production. However, very few cell lines can grow in suspension naturally, and the typical commercially used suspension cell culture systems are all mammalian in origin, making them not suitable for a lot of aquatic cases. To meet the needs of aquatic industries, a new ES culture system of aquatic origin was developed through explant outgrowth and enzyme-digesting passaging. The ES cells can adapt to suspension growth naturally and conduct transformation reversibly between the adherent and suspension modes. This culture model was confirmed to be susceptible to different fatal aquatic viruses and suitable for gene engineering as well. As a suspension culture system with no need for microcarriers and special additives, ES cells may be useful for the development of new aquatic serum-free production systems in a bioreactor or become a promising tool for the aquatic vaccine industry.

## 1. Introduction

Cells cultivated in suspension play an important role in the massive production of vaccines, biopharmaceuticals, and cell-based foods [1,2,3,4]. However, only a few mammalian cell lines currently meet the industrial requirements of stable growth in suspension, including Baby Hamster Kidney (BHK) cells, Chinese Hamster Ovary (CHO) cells, Human Embryonic Kidney 293 (HEK293) cells, and Madin Canine Kidney (MDCK) cells [5,6,7,8]. To develop more suspension culture models, various methods have been attempted to enable adherent cells to grow in suspension since the 1960s, such as using a special culture medium, long-term adaptation/domestication, or genetic engineering [9,10,11,12,13]. As for aquatic research, there is no aquatic cell line considered as versatile as CHO or Vero simian kidney epithelial (Vero) cells in the vaccine industry [14,15,16]. Because almost all known teleost somatic cell lines are adherent [17] and mammalian suspension culture systems are insensitive to many aquatic viruses, the absence of suspension culture models of aquatic origin has facilitated the application of some adherent cell lines in aquatic vaccine production, although with relatively lower efficiency. For example, Epithelioma Popuasum cuprini (EPC) cells are commonly used in spring viremia of carp virus (SVCV) vaccines [18,19], and Mandarin Fry fin type-I (MFF-1) cells were used in the development of Infectious Spleen and Kidney Necrosis Virus (ISKNV) vaccines [20]. To increase the efficiency and scale of the manufacturing of aquatic vaccines and other cell-based products, the development of an aquatic-origin suspension culture system is urgently needed.

Many freshwater eels of Anguillidae are wildly cultured as species for food consumption, and East Asia is the center of the global eel industry. In the late 1990s and the first decade of this century, *Anguilla anguilla* was the most productive species of the eel culture industry in China [21]. In 2009, *A. anguilla* was listed as CR (critically endangered) by the International Union for Conservation of Nature (IUCN), and *A. anguilla* has ceased to be the leading species of the Chinese eel industry since an exportation prohibition of glass eels took effect in 2010. Studies on *A. anguilla* cell lines are worthwhile to enhance the protection of stock and prevention of diseases in all Anguillidae members. The three most important viral pathogens of *A. anguilla* are eel virus European (EVE), eel virus European X (EVEX), and Anguillid herpesvirus (AngHV), all of which are commonly detected in the skin of infected eels and cause festering. Furthermore, these viruses have been reported to infect other aquaculture Anguillidae species, such as *Anguilla japonica* and *Anguilla rostrata* [21,22,23], implying that an *A. anguilla* skin cell line could be applied in the manufacturing of vaccines from various Anguillidae species. There were several cell lines that originated from Anguillidae species before 2006, and none of them were created from *A. anguilla* [24]. In 2007, our lab began a series of research studies on cell and tissue culture from the target organs of eels, including the kidney, liver, fin, skin and muscles, and several *A. anguilla* cell lines have been established successfully to form a small, single-species-derived cell bank [24,25,26,27].

In the present study, to build an efficient in vitro culture model for virus isolation and vaccine development using *Anguilla* sp., a new *A. anguilla* skin cell line, ES, was established and characterized. A long-term time-course growth observation was conducted using batch culture and simulated semi-continuous culture processes to determine the suspension features of the ES cell line, demonstrating its advantages as a natural reversible adherent–suspension culture model. The infection tests confirmed its susceptibility to both AngHV and red-spotted grouper nervous necrosis virus (RGNNV), two important aquatic viruses in the Chinese aquaculture industry. The ES cell line shows potential for the production of aquatic vaccines and other cell-based protein products.

## 2. Materials and Methods

### 2.1. Cell Line Development and Authentication

#### 2.1.1. Primary Culture and Subculture

The explant outgrowth method was applied to initiate the primary culture [28]. Healthy *A.anguilla* elvers with an average weight of 2 g were over-anesthetized using 20 mg/L of 3-aminobenzoic acid ethyl ester methane sulfonate (Sigma-Aldrich, St. Louis, MO, USA) and then disinfected 3 times with 2% iodine tincture and 75% alcohol. The mucus layer was removed entirely. Fresh skin tissues were excised and cut into 1 mm^3^ pieces, rinsed with ice-cold 0.01 mol/L PBS containing 200 IU·mL^−1^ of penicillin (Sangon, Shanghai, China) and 200 μg·mL^−1^ of streptomycin (Sangon, Shanghai, China) 3 times, and then attached to the inner face of a 25 cm^2^ culture flask (Corning Inc., Corning, NY, USA) coated with Leibovitz’s L-15 medium (Basalmedia, Shanghai, China). The primary culture was carried out at 26 °C with 5 mL of L-15 containing 15% fetal bovine serum (Oricell, Guangzhou, China) and antibiotics, among which 2 mL was applied after attachment, while the other 3 mL was added in 3 batches equally during 72 h. A full medium change was carried out every 72 h. After 20 d in vitro, the primary culture was digested with a 0.25% trypsin-0.02% EDTA solution (Procell, Wuhan, China) at 26 °C for 5 min and then centrifuged for 3 min at 1500 rpm. The cell suspension was prepared with 5 mL of the full culture medium and inoculated into a new 25 cm^2^ culture flask incubated at 26 °C until a confluent cell monolayer was observed.

In accordance with our previous studies [24,27], 50 continuous subcultures were carried out with a split ratio of 1:2 to build a cell line, which was characterized to grow in suspension naturally at higher passages. For cells of different passages, the subculture conditions are listed in Table 1; in particular, cells grown in suspension should be collected along with the supernatant and purified via an extra centrifugation step before digestion. For cryopreservation, the *A. anguilla* skin cells were harvested and diluted to a density of 10^6^ cells·mL^−1^ following 24 to 48 h of growth in vitro and stored in liquid nitrogen (−196 °C) through a routine programmed cooling procedure. The cryopreservation medium was made up of 70% L-15, 20% FBS, and 10% dimethyl sulfoxide (DMSO) (Sigma-Aldrich, St. Louis, MO, USA). Four weeks post cryopreservation, a recovery–viability examination was conducted using the trypan blue method [29].

#### 2.1.2. 18S rRNA Sequence Analysis

The origin of the ES cells was authenticated using 18s recombinant (r) RNA gene sequencing [30]. The total genomic DNA of passage 53 ES cells was extracted using the FastPure Blood/Cell/Tissue/Bacteria DNA Isolation Mini Kit (Vazyme, Nanjing, China). A pair of specific primers, namely, *18S-F/R* (Table 2), was designed according to the published total *A. anguilla* 18s rRNA sequence (GenBank accession No. FM946070.1) [31]. The polymerase chain reaction (PCR) amplification system was composed of a 50 μL reaction mix containing 25.0 μL of Taq Master Mix (Vazyme, Nanjing, China), 2.0 μL of each primer (10 μM each), and 1 μL of DNA templates. The PCR conditions were as follows: initial denaturation at 94 °C for 3 min, 30 cycles at 94 °C for 30 s, 55 °C for 1.5 min, and 72 °C for 1 min, with a final elongation at 72 °C for 10 min. The PCR products were analyzed via agarose electrophoresis and a gel imaging system (Thermo Nanodrop 2000, Waltham, MA, USA). The recovered DNA products were purified and sequenced by Sangon Biotech Co., Ltd. (Shanghai, China).

#### 2.1.3. Chromosome Analysis

According to our previous research [24], following 36 h in vitro, passage 50 ES cells were treated with 1 μg·mL^−1^ colchicine (Sigma-Aldrich, St. Louis, MO, USA) for 5 h. The cells were then harvested and suspended in 0.3% KCl solution. After the hypotonic treatment for 25 min, a 3-step fixation procedure with Carnoy’s fixative solution (methanol/acetic acid = 3:1, 0 °C) was conducted for 5 min at room temperature, 10 min at room temperature, and overnight at 4 °C. The fixed cells were dripped on ice-cold slides and air dried, and they were then stained with 10% Giemsa. The karyotypes of 100 metaphase cells were photographed (Eclipse TE2000-S, Nikon Instruments Inc., Tokyo, Japan) and analyzed following a classic protocol [32].

#### 2.1.4. Immuno-Cytochemical Identification

To identify the cell type of the ES cells, an immunofluorescent analysis was carried out [31]. The ES cells at passage 55 were planted on cover slips (Corning Inc., Corning, NY, USA) and transferred to a 6-well plate for 24 h at a density of 2 × 10^5^ cells·mL^−1^ and fixed with 4% paraformaldehyde (Servicebio, Wuhan, China) overnight at 4 °C before staining. The antibodies used are listed in Table 3, and the cell nuclei were marked with DAPI (Servicebio, Wuhan, China). The labeled cells were observed using an ECLIPSE TI confocal microscope (Nikon Instruments Inc., Tokyo, Japan).

### 2.2. Growth Characterization

A long-term observation of the growth characteristics of the ES cells was carried out in a batch culture process. Passage 54 ES cells were inoculated into 250 mL culture flasks at a density of 5 × 10^5^ cells·mL^−1^ and incubated in L-15 medium supplemented with 10% FBS. Continuous micrography was conducted for 16 d (Eclipse TE2000-S, Nikon, Nikon Instruments Inc., Tokyo, Japan). At 4 d, 8 d, and 12 d following inoculation, the suspension ratio and the major axis of the cell clusters were measured and compared; the cell clusters in suspension were finally collected and subcultured into new flasks at the same density after thorough digestion (Table 1). Another 16 d observation was carried out in a simulated semi-continuous culture process. At 4 d, 8 d, and 12 d following inoculation, the cells in suspension were removed and accompanied by a full medium change, and the secondary growth of the adherent parts was then recorded and evaluated.

The growth curves of the ES cells were plotted at passage 55. With an interval of 24 h, the average adherent cell density and suspension cell density of different groups were measured separately using a hemocytometer (*n* = 3). The cell population doubling time (PDT) was determined using the formula below (where *T* means the population doubling time of the logarithmic phase, *N_t_* represents the final number of cells, *N*_0_ represents the initial number of cells, and *t* represents the time interval between *N_t_* and *N*_0_) [33] as follows:*T* = *t* [lg2 (lg *N_t_* − lg *N*_0_)^−1^].(1)

The purified ES cells were diluted to two densities of 2 × 10^5^ cells·mL^−1^ and 10^6^ cells·mL^−1^ and then inoculated into 50 mL flasks and calculated. The serial tests based on the serum percentage and temperature differences were blocked in the 2 × 10^5^ cells·mL^−1^ group due to the absence of suspension cells (26 °C, 10% FBS). The 10^6^ cells·mL^−1^ samples were incubated and measured as follows: at 26 °C, five groups of ES cells were incubated in L-15 medium supplemented with 3%, 5%, 8%, 10%, and 15% FBS; subsequently, the second batch of five groups was kept in L-15 medium containing 8% FBS and incubated at 15 °C, 20 °C, 26 °C, 30 °C, and 37 °C; and finally, the growth curves of the adherent and suspension cells were plotted. The ratios between the suspension cell numbers and the total cell numbers (%) of each group at 72 h, 120 h, and 168 h were analyzed and compared.

### 2.3. Cell Transfection

The ES cells at passage 50 were inoculated in a 6-well culture plate (Corning Inc., Corning, NY, USA) with a density of 10^6^ cells·mL^−1^. After being incubated at 26 °C for 24 h, the cells were transfected with the GFP reporter gene carrier pEGFP-N1 (Biomed, Beijing, China) using a lipofectamine™ 3000 kit (Invitrogen, Carlsbad, CA, USA). According to the official protocol with some modifications, the working diluent contained 2.5 μg of pEGFP-N1 plasmid, 5 μL of lipofectamine™ 3000 reagent, and 250 μL of minimum essential medium (MEM) (Gibco, New York City, NY, USA). The transfected cells were observed and visualized via fluorescence microscopy (Eclipse TE2000-S, Nikon, Nikon Instruments Inc., Tokyo, Japan) every 24 h, and the transfection efficiency was evaluated based on the ratio of green fluorescence-positive cells to all cells in 10 distinct optical fields.

### 2.4. Susceptibility Test

The ES cells were infected with 2 aquatic viruses, namely, RGNNV and AngHV. The ES cells (passage 50) inoculated at a density of 5 × 10^5^ cells·mL^−1^ were first incubated at 26 °C for 18–24 h to form a confluent monolayer and then transferred to the maintenance medium (L-15 containing 3% FBS) at 30 °C at 4 h prior to infection. The viral samples were prepared with the supernatant from RGNNV-infected EPC cells and AngHV-infected EL cells [24,34] and then added to the ES cells (200 μL per flask). The infected cells were observed under a phase-contrast microscope (Eclipse TE2000-S, Nikon, Nikon Instruments Inc., Tokyo, Japan) continuously to access cytopathic effects (CPEs). A subculture was carried out every 7 days or after reaching a cytopathic rate of over 60%. The infected cell number was estimated by counting the living cells using a flow cytometer (BD Accuri™, Becton, Dickinson and Co., Franklin Lakes, NJ, USA) and compared with the control group.

After 3 subcultures, total DNA was extracted as PCR templates from the infected cells using a FastPure Blood/Cell/Tissue/Bacteria DNA Isolation Mini Kit (Vazyme, Nanjing, China) at 96 h post-infection. The PCR detection of RGNNV was conducted with a pair of specific primers, namely, *VNN F2/R3* (Table 2) [35], while identification for AngHV was conducted with two pairs of specific primers, namely, *HVAF/R* and *HVApolF/polR* (Table 2) [36,37].

### 2.5. Data Analysis

Each experiment was repeated at least three times. The data are shown as mean ± standard deviation (SD), and statistical significance was determined via multiple comparisons within the framework of one-way analysis of variance (ANOVA, Turkey’s HSD test) [33] using GraphPad Prism 9.5 (www.graphpad.com, accessed on 31 July 2024).

### 2.6. Electron Microscopy

At 48 h post-infection, the infected ES cells were pre-fixed with 2.5% glutaraldehyde in cacodylate buffer (0.1 M, pH 7.4) for 1 min and then harvested and suspended in 2.5% glutaraldehyde for 4 h at room temperature (25–28 °C). After centrifugation at 1000 rpm for 2 min, the precipitate was incubated in 2.5% glutaraldehyde overnight at 4 °C [38]. Ultra-thin sectioning and transmission electron microscopy (TEM) were subsequently conducted by Servicebio Technology Co., Ltd. (Wuhan, China).

## 3. Results

### 3.1. Cell Line Establishment and Authentication

#### 3.1.1. Primary Cell Culture and Cell Line Establishment

Following inoculation, the cells migrated outwards from the explants within 48 h, and radial outgrowths covered over 70% of the area of the flask’s inner surface in 2 weeks (Figure 1a). After the first digestion, subcultures were performed at an interval of approximately 24 h continuously (split ratio = 1:2). At passage 20, the cell strain was purified and demonstrated fibroblast-like fully adherent cells (Figure 1b). The eel skin (ES) cell line was subcultured over 60 times in vitro and usually maintained in L-15 containing 8% FBS at 26 °C (Figure 1c). The ES cells recovered from liquid nitrogen storage at the 60th subculture reached confluency within 3 days; their average viability was evaluated to be over 80% (Figure 1d).

#### 3.1.2. 18S rRNA Sequence Analysis

The origin of the ES cells was authenticated via 18S rRNA gene analysis. Using the specific pair of primers *18S-F/R* (Table 2), an expected product of 1714 bp was amplified and sequenced, which was 100% identical to the published *A. anguilla* 18S rRNA sequence (GenBank: FM946070.1). These results confirmed that the ES cells derived from *A. anguilla* were kept uncontaminated during passaging.

#### 3.1.3. Chromosome Analysis

A total of 100 metaphase ES cells (passage 50) were analyzed; their chromosome number ranged from 18 to 96 (Figure 2a). The modal diploid karyotype of 2*n* = 38 was presented in only 15% of all samples, suggesting that the ES cell line was aneuploid (Figure 2b) [39].

#### 3.1.4. Immuno-Cytochemical Identification

To determine the nature of the ES cells, five kinds of cell markers were identified with six monoclonal antibodies, as described in Table 3. The fluorescent signals were observed to be strongly positive for the vimentin and desmin groups (Figure 3a,b), moderately positive for the fibronectin and collagen-1 group (Figure 3d,e), and negative for the cytokeratin group (Figure 3c). Vimentin and desmin were globally expressed in the cytoplasm (Figure 3a,b); however, fibronectin and collagen-1 were distributed at the cell surface with punctate agglomerations (Figure 3d,e). These results suggested that the ES cell line was fibroblastoid [40].

### 3.2. Growth Characterization

#### 3.2.1. Batch Culture Process and Simulated Semi-Continuous Culture Process

In the batch culture process with an inoculum density of 5 × 10^5^ cells·mL^−1^, a confluent monolayer was formed by day 2; spheroidal cell clusters were first observed to be adherent and then automatically turned into suspension over time with volume increases. In the meantime, co-localized attraction and disintegration of the adherent cell layer occurred along with the presence of suspending cell clusters (Figure 4a). The suspension ratio and volume change in the cell clusters at different stages were measured and analyzed (Figure 4e). On day 4, all cell clusters were attached to the surface of the cell layer, with their major axis ranging from 48.42 to 81.09 μm; on day 8, the adherent and suspension clusters shared a major axis ranging from 127.22 to 221.75 μm; and by day 12, all clusters observed were in suspension with a major axis in the range of 135.38–355.04 μm, while the adherent cells were still observed to be in a dispersed pattern (Figure 4a,e).

For the cell clusters that were collected and thoroughly digested, the natural process from being completely adherent to cluster formation was reproduced exactly within 96 h post-re-inoculation (Figure 4b), and subsequent observations confirmed the replication of an autonomous suspension. These results suggested that the ES cells can form a biphasic culture system, switching between the adherent and suspension modes reversibly (Figure 4, Appendix A).

In the semi-continuous culture process simulated in the flasks, at the time points of day 4 and day 8 after all culture medium and suspension cells were removed (Figure 4c), the adherent cells succeeded in compensating the lesion inflicted by cluster formation and reconstructed a confluent cell layer (Figure 4d); at the time point of day 12, the new cells were overwhelmingly agglomerated in suspension (Figure 4c,d). These results indicated the potential of ES cells to reach higher production limits than those obtained using a simple batch culture process.

#### 3.2.2. Growth Characteristics

When inoculated at a low density of 2 × 10^5^ cells·mL^−1^, the suspension growth of the ES cells was inhibited significantly, and the total growth curve and the adherent growth curve almost coincided. From 48 to 120 h following inoculation, the ES cells (passage 55) were found in the logarithmic phase with a PDT of 35.46 h (Appendix A).

At an inoculum density of 10^6^ cells·mL^−1^, the ES cells became completely attached and formed a confluent monolayer at all five serum percentages tested (Appendix A). The highest growth rate, the ratio of suspension cells to the total number of cells, and the number of cell clusters in suspension were all positively correlated with the serum percentage (Figure 5a,c and Appendix A). In particular, there were suspension cells, which dispersed in the 3% group and formed no clusters (Appendix A). From 48 to 144 h after inoculation, the adherent cells were observed to first move into the logarithmic growth phase and then form a second growth peak following the increase in the suspension growth rate (Figure 5a), suggesting probable compensatory effects for the cell-layer lesions inflicted by the transposition of cell clusters into suspension; in the groups subjected to high FBS percentages (8%, 10% and 15%), the suspension cells surpassed the adherent cells to be the major population in vitro (Figure 5a and Appendix A). The highest adherent growth rate was measured in the 8% group, while the maximum suspension growth rate was recorded in the 10% group (Figure 5a,c).

When inoculated at a density of 10^6^ cells·mL^−1^, the ES cells became fully attached and continued being highly productive at a temperature range from 15 to 37 °C. At 15 °C, a delayed attachment at 48 h was observed, and the subsequent cell growth pattern was network like (Appendix A); at the other four temperatures tested, the ES cells finished attachment within 24 h and succeeded in forming a confluent monolayer. Suspension cell clusters were observed only in the three hypothermal–mesothermal groups (15 °C, 20 °C, and 26 °C) (Appendix A), while all suspension cells were presented in a dispersed pattern at 30 °C and 37 °C (Appendix A). A significant increase in cell volume was observed at 37 °C (Appendix A). The logarithmic growth phase was plotted within the interval of 48 to 144 h following inoculation; the highest adherent growth rate was measured in the 30 °C group, while the maximum suspension growth rate was recorded in the 20 °C group (Figure 5b,c and Appendix A).

In the serum difference tests, at 72 h after inoculation, the suspension ratio (%) of the cells showed no significant difference among the five groups; at 120 h post-inoculation, the only significant difference was observed between the 3% and 5% groups; and at 168 h after inoculation, the suspension ratios of the 3% and 5% groups were significantly lower than those of the 8%, 10%, and 15% groups, while the suspension ratio of the 10% group was significantly higher than that of the 15% group (Figure 6a). In the temperature difference tests, the suspension ratio (%) of the 20 °C group was significantly higher than those of the other four groups at all time points. In addition, at 72 h after inoculation, the suspension ratio (%) of the 30 °C group was significantly higher than that of the 37 °C group; at 120 h post-inoculation, the suspension ratio (%) of the 15 °C group was significantly lower than those of the 26 °C, 30 °C, and 37 °C groups; and at 168 h after inoculation, the suspension ratio of the 30 °C group was significantly higher than those of the 26 °C and 37 °C groups (Figure 6b).

From these results, we optimized the subculture conditions for the ES cells as follows: a temperature of 26 °C and 8% FBS (PDT = 29.81 h) and an inoculum density higher than 5 × 10^5^ cells·mL^−1^ would be essential if suspension growth was expected. The maintenance culture conditions for virus infection were optimized as follows: a temperature of 30 °C and 3% FBS, with which the ES cells could form a confluent cell layer within 24 h and keep it for over 14 d.

### 3.3. Cell Transfection

Green fluorescent signals were first detected at 24 h following transfection and lasted for at least 96 h. At 48 h post-transfection, the transfection efficiency of pEGFP-N1 was measured to be about 10% (Figure 7).

### 3.4. Susceptibility Tests

Both the AngHV and RGNNV samples were able to cause CPEs on the ES cells continuously across the three subcultures. Compared with the confluent control groups (Figure 8a,c), the AngHV-infected cells were observed to have become rounded and detached from the monolayer at 60 hpi (hours post-infection) (Figure 8b), while the RGNNV-infected cells were observed to have shrunk and collapsed massively, accompanied by an increase in mortality. At 72 hpi, the mortality rate of the RGNNV-infected group was evaluated to be over 85% (Figure 8d).

The particles of both viruses were determined and found in the cytoplasm using an electron microscopy assay. Viral factories (VFs) with a lower electron density than that of the surrounding cytoplasm were present in the AngHV-infected cells [41], sheltering numerous empty capsids and immature viral particles inside, while mature virions were localized outside (Figure 8e). As for the RGNNV-infected cells, viral particles of different ages were all found dispersed in the cytoplasm (Figure 8f).

Following PCR amplification of the AngHV *pol* gene, the expected 390 and 622 bp products were readily detected (Appendix A) [42]; for the RGNNV-positive samples, a 420–431 bp product was also detected via a standard viral nervous necrosis (VNN) diagnostic assay (Appendix A) [35].

## 4. Discussion

In the last 20 years, fish cell culture has been weaning itself off the heritage of mammalian techniques. Over 90% of new cell lines originated from fish have turned to wide-temperature CO_2_-free culture models based on Leibovitz’s L-15 medium, which has abolished the ventilation system and made the culture procedure more compact and cost effective [17,43,44]. Furthermore, the high recovery ratio of ES cells cryopreserved in a single-cell suspension has been authenticated (Figure 4), while rapid freezing–thawing is also used in in vitro adaptation of stem cell differentiation and organoid induction [45,46,47]. Developing a cryopreservation procedure for ES cells as suspension clusters may be a viable idea to increase productivity further. Therefore, a suspension cell culture model from fish represents a combination of a stable environment, high cell density, and easy expansion.

The ES cells were found to induce a naturally conditioned suspension growth pattern in the batch culture process, which could be modulated by adjusting routine culture parameters, including initial inoculum density, serum percentage, and temperature. To optimize the subculture protocol, both the adherent and suspension applications were taken into consideration. According to Table 1, the 96–168 h interval of purified suspension cells is too long for small-scale experiments; therefore, the adherent subculture methods were chosen. To observe the adherent–suspension transformation of ES cells quickly, a minimum inoculum density of 5 × 10^5^ cells·mL^−1^ is suggested (Figure 4 and Appendix A). In the re-inoculation test of the counted cells on day 7 (26 °C, 8% FBS), an inoculum density over 2 × 10^7^ cells·mL^−1^ resulted in full suspension within 96 h. The optimum density and cycle for suspension ES cells on a large scale should probably be measured with a high lower limit [3]. A low serum percentage inhibited the natural initiation of suspension growth (Figure 5 and Appendix A). Especially, in the 3% FBS batch culture process, the ES cells could be maintained for over 2 weeks at a density of 10^7^ cells·mL^−1^, which implied the tolerance of the ES cells to an oligotrophic environment and the further possibility for serum-free culture adaptation [48]. At all time points analyzed, the active suspension growth of the ES cells at 20 °C as cell clusters was remarkable (Figure 5 and Figure 6b). In their very early life stages with a culture temperature limit of 16–22 °C, the expression patterns of the *A. anguilla* larval genes *gh* and *igf-1* were reported to be delayed at 16 °C but accelerated at 20–22 °C [49]. Investigating the regulation of these genes may help in understanding the selective preference of suspension ES cells for a certain temperature window. In all batch culture tests, the PDT of the suspension cells was measured to be shorter than that of the adherent cells from the same sample (Figure 5c), suggesting the potential for high biomass yield in a purified suspension culture. At 26 °C and 8% FBS, the maximum growth rate of adherent and suspension cells came to a minimum distance (Figure 5c), and the cell suspension ratio (%) presented the least significant differences among the multiple comparisons (Figure 6); from passage 55–68, the ES cells were maintained stably with this subculture protocol, confirming its convenience.

The ES cells were found to present susceptibility to multiple aquatic viruses. AngHV is the most threatening viral pathogen for Anguillidae members, and it has been found in most aquaculture areas in Asia and Europe [23,37,50,51,52,53,54,55], as well as in wild silver eels from the migrating populations [56]. *A. anguilla* prefers a culture temperature range of 18–26 °C, while other Anguillidae commercial species, which are all AngHV sensitive, can adapt to higher culture temperatures of up to 30 °C [21]. Since 2008, our lab has developed a series of cell lines that originated from different tissues of *A. anguilla* to study AngHV [24,25,26,27]. AngHV-infected syndromes are mainly shown in the skin, gills, kidney, liver, and spleen. Compared with previous infection samples at 20–26 °C using viscera cell lines [24,57], the ES infection model presented typical CPEs rapidly at 30 °C, which accords with the water temperature during the epidemic season in Fujian and Guangdong province, the main eel culture-producing areas in China, suggesting that the ES cell line can be a universal model for AngHV vaccine production from Anguillidae members.

In addition to AngHV, the ES cells also exhibited susceptibility to RGNNV, which was once detected in cultured *A. anguilla* in Taiwan [58], but it is not a usual pathogen of farmed *Anguilla rostrata* and *Anguilla marmorata* in recent years; its relative, the striped jack nervous necrosis virus (SJNNV), was observed being carried by wild eels of Spain [59]. At present, RGNNV is predominant worldwide, with grouper (*Epinephelus* sp.) and European sea bass (*Dicentrarchus labrax*) as the most susceptible victim species [34]. Our results implicated VNN as a threat in abeyance for cultured eels and offered a new potential culture system for RGNNV vaccine production.

As a traditionally popular economic species, *A. anguilla* is now critically endangered due to overfishing, environmental pollution, hydrological change, and diseases [21]. Being listed in Appendix II of the Convention on International Trade in Endangered Species of Wild Fauna and Flora (CITES), *A. anguilla* has conceded its leading place in the Chinese eel industry to *A. rostrata* and *A*. *marmorata* since the exportation prohibition of glass eels in 2010 [60]. Because of their special catadromous life cycle [61], the failure of artificial reproduction, and the catch-depending aquaculture mode, the abundance of many Anguillidae species in their original habitat is decreasing every year [62]. Research studies on *A. anguilla* are worthwhile to enhance the protection of stock and prevention of diseases in all Anguillidae members. Furthermore, culturing cell lines originating from edible parts, such as the skin and muscles, may bring the unique flavor of *A. anguilla* back to the table through cell-based food production via in vitro organoid induction and mass bioreactor production [63,64,65]. ES cells’ suspension feature places them well in the competition of cellular agriculture and, it is a viable tool to preserve the genetic resources of endangered eels.

## 5. Conclusions

In conclusion, an aneuploid cell line from ES (eel skin) was developed as the first aquatic culture system that can naturally adapt to suspension culture. This fibroblastic culture system can be reversibly converted between the adherent and suspension modes, and it can reach a high cell density of a magnitude of 10^7^ cells·mL^−1^ easily without the need for special additives or microcarriers. The ES cells can be transfected with foreign reporter genes and are sensitive to multiple important aquatic viruses. All these results suggest that the ES culture system is more cost effective by removing the processes of suspension domestication and carrier dissolution, which can benefit the production of aquatic vaccines, biopharmaceutics, or cellular agriculture.

## Figures and Tables

**Figure 1 biology-13-01068-f001:**
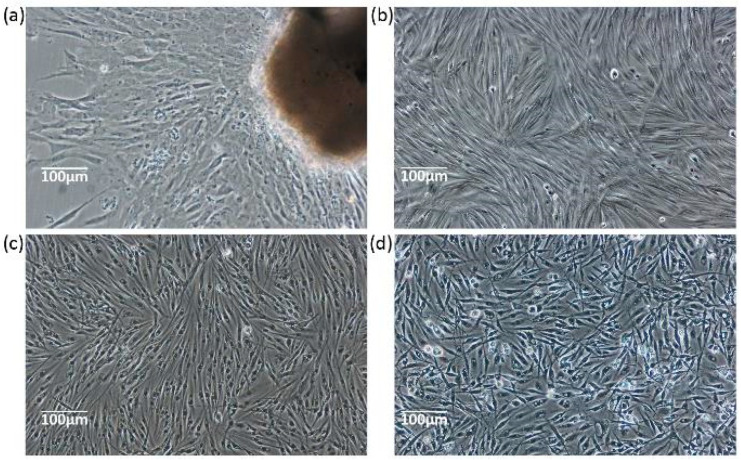
Development of the ES cell line. (**a**) Primary culture of *A. anguilla* skin; (**b**) passage 20 ES cells; (**c**) passage 55 ES cells at 24 h in vitro; and (**d**) passage 60 ES cells at 72 h post recovery from liquid nitrogen.

**Figure 2 biology-13-01068-f002:**
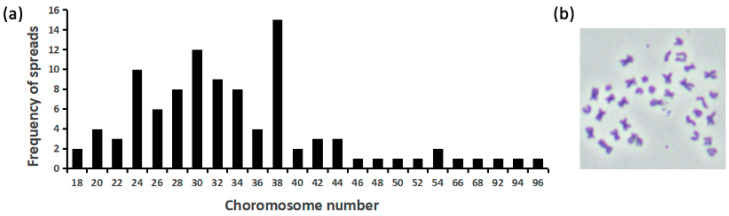
Chromosome analysis of ES cells at passage 50. (**a**) The distribution of chromosome number, among which only 15% of samples (*n* = 100) presented a diploid karyotype (2*n* = 38). (**b**) The metaphase of a typical aneuploidy ES cell.

**Figure 3 biology-13-01068-f003:**
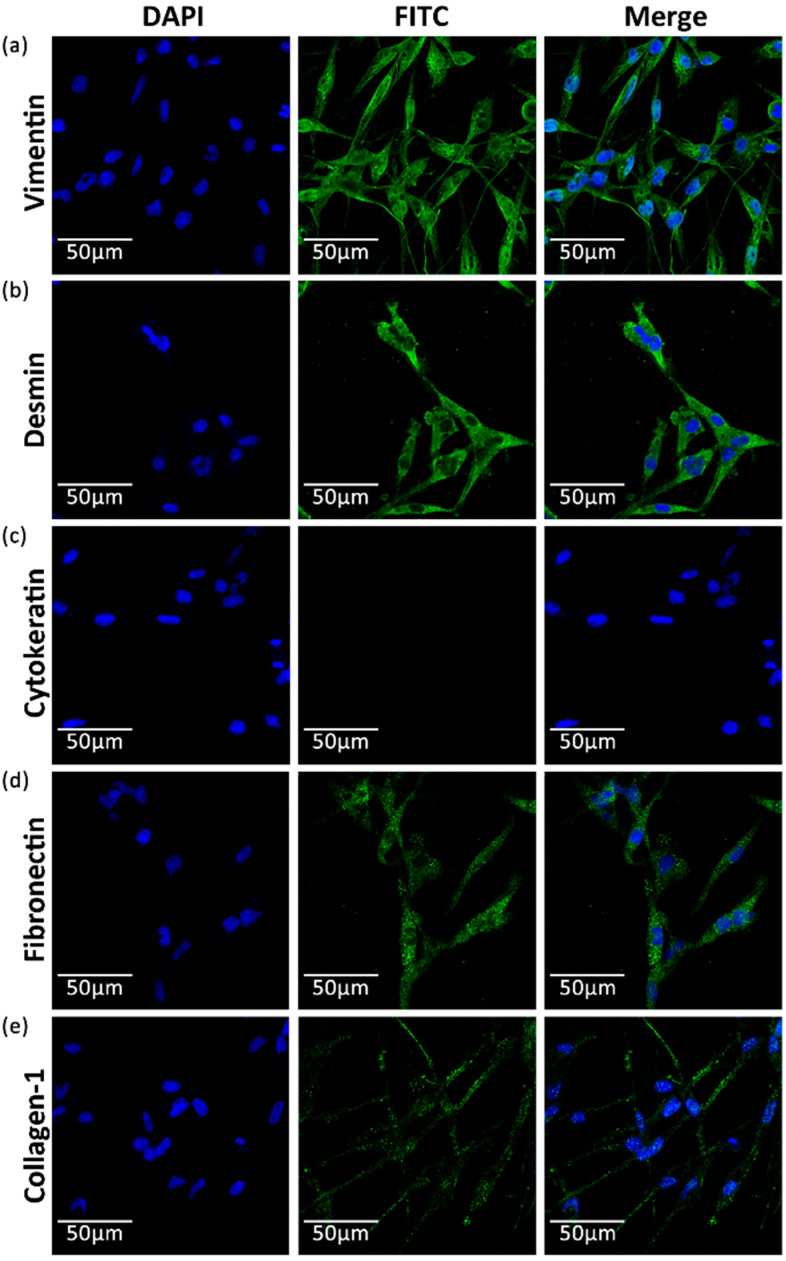
Immunofluorescence staining of 5 markers in passage 55 ES cells: (**a**) vimentin; (**b**) desmin; (**c**) cytokeratin; (**d**) fibronectin; and (**e**) collagen-1. The cell nuclei are indicated with DAPI (blue), and the cell markers are indicated with FITC (green).

**Figure 4 biology-13-01068-f004:**
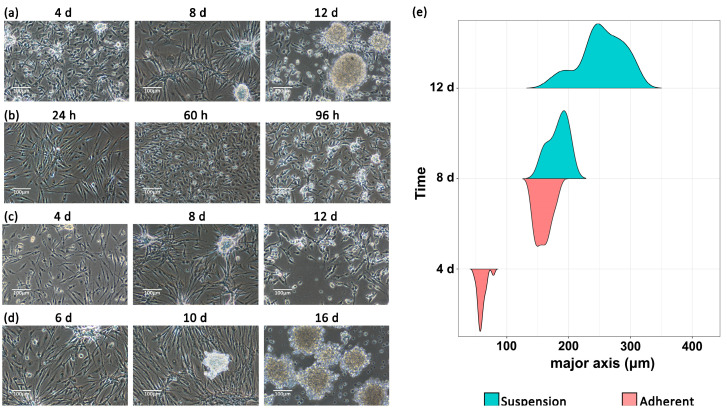
Passage 54 ES cells in a batch culture process and a simulated semi-continuous culture process. (**a**) The autonomous formation of suspension cell clusters; (**b**) the re-formation of cell clusters after digestion; (**c**) the harvest of suspension cells with a full medium change; (**d**) the compensation of adherent cell-layer lesions and output of new suspension cell clusters; (**e**) the violin plots of the adherent/suspension ratio and the major axis of the cell clusters over a 12 d long-term observation.

**Figure 5 biology-13-01068-f005:**
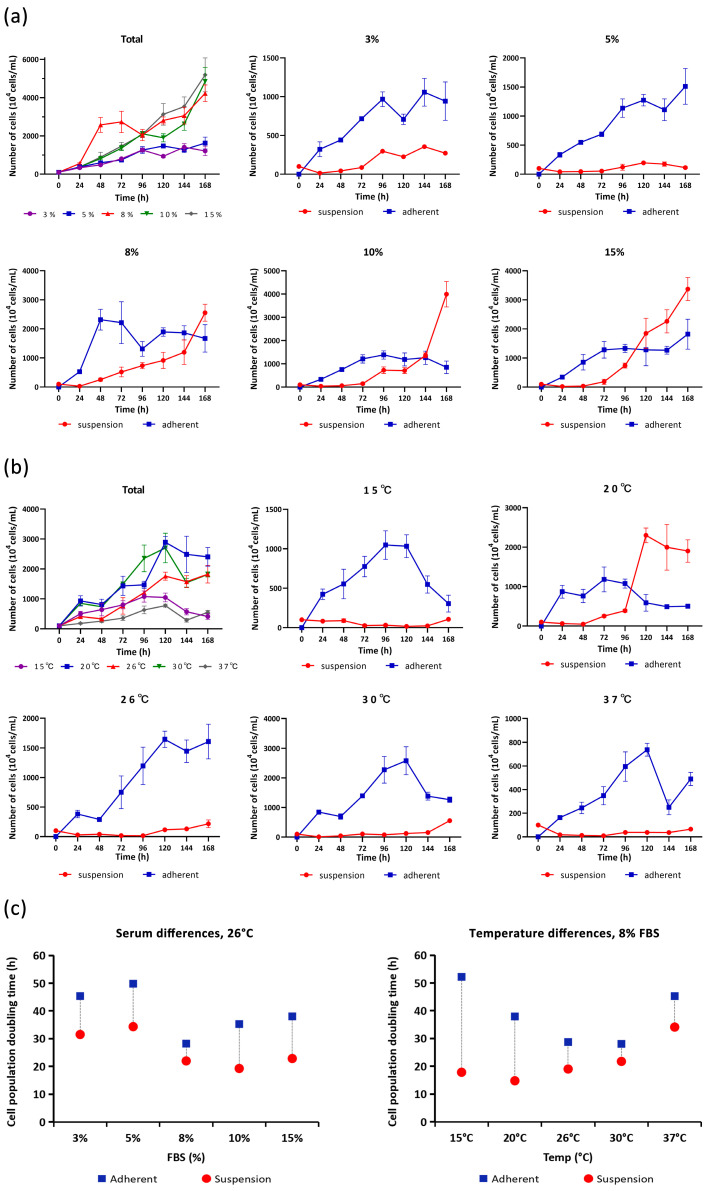
The growth characteristics of passage 55 ES cells under different serum percentages and temperatures. (**a**) Growth curves at 26 °C with different FBS%; (**b**) growth curves at different temperatures with 8% FBS (inoculum density = 10^6^ cells·mL^−1^); and (**c**) the population doubling time (PDT) of adherent and suspension cells. The growth curve values are exhibited as the mean ± standard deviation (SD), *n* = 3.

**Figure 6 biology-13-01068-f006:**
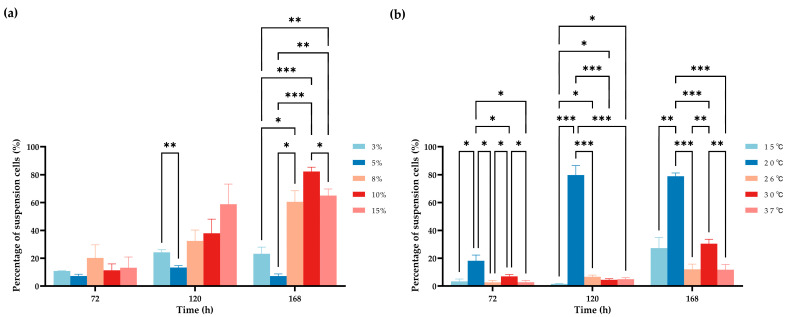
Analysis of the suspension ratio (%) of passage 55 ES cells under different serum percentages and temperatures. (**a**) Multiple comparisons at 26 °C with different FBS%; (**b**) multiple comparisons at different temperatures with 8% FBS (inoculum density = 10^6^ cells·mL^−1^). * *p* < 0.05, ** *p* < 0.01 and *** *p* < 0.001*; n* = 3.

**Figure 7 biology-13-01068-f007:**
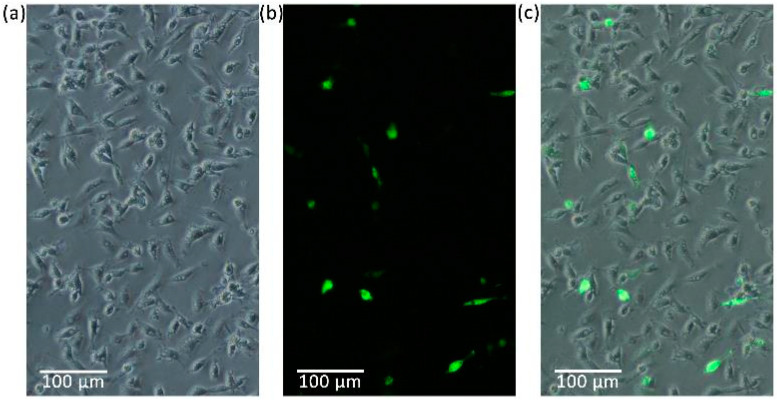
Passage 50 ES cells transfected with pEGFP-N1 for 48 h. (**a**) An optical micrograph; (**b**) a green fluorescence micrograph; and (**c**) a merging of the optical and GFP graphs.

**Figure 8 biology-13-01068-f008:**
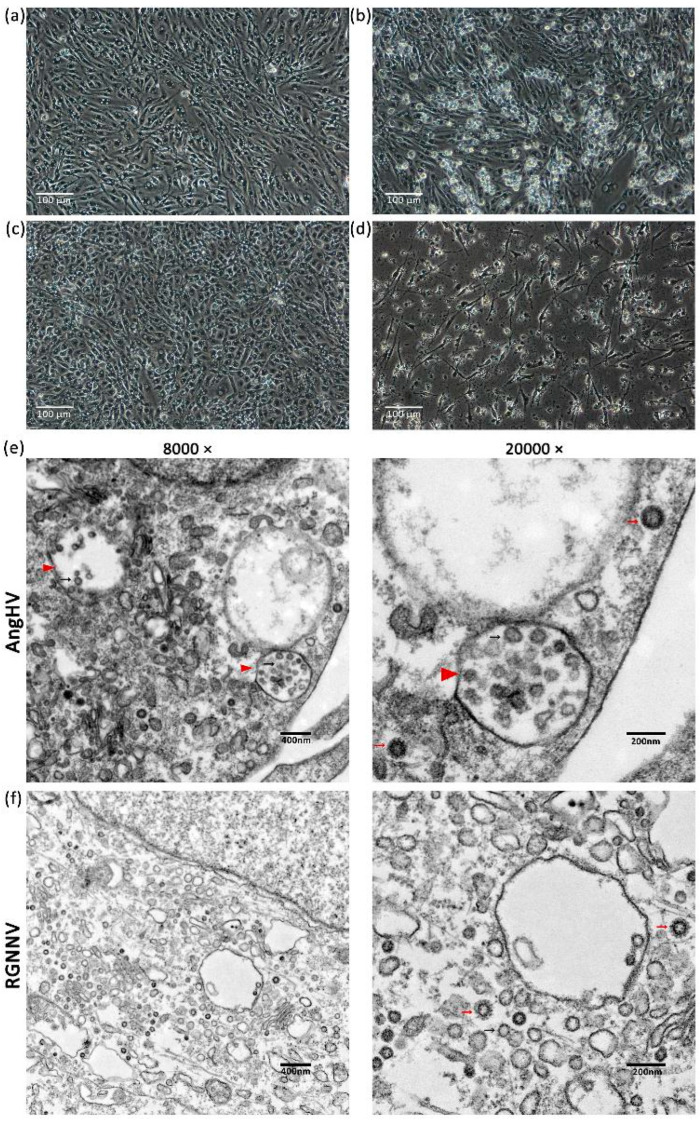
Susceptibility tests of passage 54 ES cells to two aquatic viruses. (**a**) The control group at 60 h in vitro; (**b**) ES cells infected with Anguillid herpesvirus (AngHV), 3 subcultures at 60 hpi; (**c**) the control group at 72 h in vitro; (**d**) ES cells infected with red-spotted grouper nervous necrosis virus (RGNNV), 3 subcultures at 72 hpi; (**e**) an electron microscopic image of AngHV particles inside infected ES cells at 48 h post-infection; (**f**) an electron microscopic image of RGNNV particles inside infected ES cells at 48 h post-infection. Red triangles indicate viral factories (VFs) with a low electron density; small red arrows indicate mature viral particles; small black arrows indicate immature viral particles.

**Table 1 biology-13-01068-t001:** The subculture parameters of eel skin (ES) cells.

Cell State [A/S]	Cell Passage [d]	Digestion Time [s]	Centrifuge Speed [rpm]	Centrifuge Time [min]	FBS [%]	Interval [h]
Adherent	2–20	60	1500	3	15	24–48
Adherent	21–50	30–45	1800	5	10	24
Adherent	>50	30–45	1800	5	8	24
Suspension	>50	300	1800	5	8	96–168

**Table 2 biology-13-01068-t002:** The PCR primers used in this study.

Primer Name	Forward Sequence (5′-3′)	Reverse Sequence (5′-3′)
*18s-F* */R*	TATGCTTGTCTCAAAGATTAAGCCATGC	CACCTACGGAAACCTTGTTACGA
*HVA F* */R*	TTGAGGTTGTTGTCGTGCC	CTCTCATGTCATCCAGACGG
*HVA pol F* */R*	GTGTCGGGCCTTTGTGGTGA	GTGTCGGGCCTTTGTGGTGA
*VNN F2* */R3*	CGTGTCAGTCATGTGTCGCT	CGAGTCAACACGGGTGAAGA

**Table 3 biology-13-01068-t003:** Antibodies for immunofluorescent identification.

Antibody	Type	Source
Acidic cytokeratin	Monoclonal mouse	Servicebio, Wuhan, China
Basic cytokeratin	Monoclonal mouse	Servicebio, Wuhan, China
Collagen-1	Monoclonal mouse	Servicebio, Wuhan, China
Desmin	Monoclonal mouse	Servicebio, Wuhan, China
Fibronectin	Monoclonal mouse	Servicebio, Wuhan, China
Vimentin	Monoclonal mouse	Servicebio, Wuhan, China
Fluorescein isothiocyanate (FITC) conjugated	Goat anti-mouse IgG	Servicebio, Wuhan, China

## Data Availability

The data that support the findings of this study are available from the corresponding author upon reasonable request.

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
