# Peer review of "Novel Eel Skin Fibroblast Cell Line: Bridging Adherent and Suspension Growth for Aquatic Applications Including Virus Susceptibility"

_biology, 2024, doi:10.3390/biology13121068_

Round 1

Reviewer 1 Report

Comments and Suggestions for Authors This manuscript describes the development and analytic description of an aneuploid fibroblastic cell line derived from the primary culture of the skin of the eel Anguilla anguilla.  The properties of this cell line allow the culture of either suspended or attached cells depending on varied growth conditions, such as temperature and culture media.  These properties could be useful for further studies leading to the development of vaccines and cultured, edible, fish-derived protein. The experiments are carefully controlled and the results clearly presented in the provided figures.  The listed references are appropriate.

This work provides a technical advancement to the field.   Cell culture systems are of practical importance for the future development of vaccines, which themselves are critical for the advancement of aquaculture.  The experimental evidence and characterization of this new cell line supports the authors' conclusion that this cell culture system will be useful to other researchers in eel aquaculture.   Table 1 (line 103) Under Cell Passage column need to add "[d]" Comments on the Quality of English Language

Although the major context of the writing is understandable, there are numerous grammatical errors throughout the manuscript that need to be corrected before publication.  As examples (but not inclusive of all the errors in the manuscript):

lines 346-347:  "...were watched as shrank."

lines 400-401: "....wild eels in Spanish."

line 405:  "As a centuried popular economic species...."

Author Response

Comments 1: Comments and Suggestions for Authors

This manuscript describes the development and analytic description of an aneuploid fibroblastic cell line derived from the primary culture of the skin of the eel Anguilla anguilla.  The properties of this cell line allow the culture of either suspended or attached cells depending on varied growth conditions, such as temperature and culture media.  These properties could be useful for further studies leading to the development of vaccines and cultured, edible, fish-derived protein. The experiments are carefully controlled and the results clearly presented in the provided figures.  The listed references are appropriate. 

This work provides a technical advancement to the field.   Cell culture systems are of practical importance for the future development of vaccines, which themselves are critical for the advancement of aquaculture.  The experimental evidence and characterization of this new cell line supports the authors' conclusion that this cell culture system will be useful to other researchers in eel aquaculture.   Table 1 (line 103) Under Cell Passage column need to add "[d]"

Response 1: Thank you for your pertinent comments, we agree with this comment. The Table 1 has been updated as suggested.

Comments on the Quality of English Language

Although the major context of the writing is understandable, there are numerous grammatical errors throughout the manuscript that need to be corrected before publication.  As examples (but not inclusive of all the errors in the manuscript):

lines 346-347:  "...were watched as shrank."

lines 400-401: "....wild eels in Spanish."

line 405:  "As a centuried popular economic species...."

Response 2: Agree. We’ve applied for a professional rapid language editing service of MDPI Author Service before the revision was submitted, and all these grammatical errors have been corrected.

Appendix

Supplemented References:

  1. Haenen, O.; van Ginneken, V.; Engelsma, M.; van den Thillart, G. Impact of Eel Viruses on Recruitment of European Eel. In ed.; van den Thillart, G.; Dufour, S.; Rankin, J.C.,Eds.; Springer Netherlands: Dordrecht, 2009; pp. 387-400. doi: 10.1007/978-1-4020-9095-0_16
  2. Bryson, S.P.; Joyce, E.M.; Martell, D.J.; Lee, L.E.; Holt, S.E.; Kales, S.C.; Fujiki, K.; Dixon, B.; Bols, N.C. A Cell Line (HEW) from Embryos of Haddock (Melanogrammus aeglefinius) and its Capacity to Tolerate Environmental Extremes. Mar Biotechnol (NY) 2006, 8, 641-653. doi:10.1007/s10126-005-6163-1
  3. Zhao, Z.; Li, F.; Ning, J.; Peng, R.; Shang, J.; Liu, H.; Shang, M.; Bao, X.Q.; Zhang, D. Novel Compound FLZ Alleviates Rotenone-Induced PD Mouse Modelby Suppressing TLR4/MyD88/NF-kappaB Pathway through Microbiota-Gut-Brain Axis. Acta Pharm Sin B 2021, 11, 2859-2879. doi:10.1016/j.apsb.2021.03.020
  4. Merten, O.W. Introduction to Animal Cell Culture Technology-Past, Present and Future. Cytotechnology2006, 50, 1-7. doi:10.1007/s10616-006-9009-4
  5. Schmidt, J. Breeding Places and Migrations of the Eel. Nature1923, 111, 51-54. https://www.nature.com/articles/111051a0.pdf
  6. Arai, T. Do we Protect Freshwater Eels Or Do we Drive them to Extinction? Springerplus2014, 3, 534. doi:10.1186/2193-1801-3-534

Reviewer 2 Report

Comments and Suggestions for Authors

A brief summary:

This MS by Zheng et al. developed a reversible 2D-3D culture model originated from Anguilla anguilla skin, which can naturally adapt to adherent and suspension growth reversibly without any carrier. This cell line, named ES, has been kept in vitro continuously for over 12 months, and subcultured for over 60 times. In the chromosome analysis, the ES cell line has been proved to be heteroploid. The ES cells can adapt to a wide temperature range of 15-37°C, and keep highly proliferative at a serum percentage ranging from 3-15%; we’ve optimized the routine subculture conditions as: 26°C, 8% FBS and inoculum density higher than 5×105 cells·mL-1, and the maintenance culture conditions for virus infection have been optimized as: 30°C, 3% FBS. The green fluorescence protein (GFP) reporter gene has been successfully expressed in the ES cells. Furthermore, the ES cell line has demonstrated the susceptibility to Anguillid herpesvirus (AngHV) and Red-spotted Grouper Nervous Necrosis Virus (RGNNV), which are distantly related, suggesting its manifold possibilities for vaccine manufacture.

Overall, I thought this was a well-executed study in a system with limited previous knowledge of this level on this specific topic. I appreciated their multi-faceted approach (Molecular biology and Immunofluorescent) and the time course involved in this work and think it add significant merit to their work. I do think that their overall conclusions were related to results. Overall, I think this is good work, but should undergo some revision before acceptance.

1)    General concept comments.

I think the author should seek help with writing in English, I hope round of editing with English native language would further improve the quality of the writing. The English grammar should be checked again.

I also have a couple suggestions.

2)    Specific comments

Title “A reversible 2D-3D culture model from European eel (Anguilla anguilla) skin with susceptibility for different aquatic viruses”

“2D-3D” is not clear, I suggest it is better change “2D to “adherent”, and change “3D” to “suspension”. The author may consider better title such as “A reversible culture model between adherent and suspension from European eel (Anguilla anguilla) skin with susceptibility for different aquatic viruses”

1.     Introduction

In “Introduction”, the author introduced mainly focused on “2D” and “3D”. However, there is little information on European eel (Anguilla anguilla). “In the present study, a micro-carrier-free culture system originated from European eel (Anguilla anguilla) skin has been introduced”

Why the author chose the skin of European eel to do this study, but not other fish?

What about the aquaculture output and advantage of European eel in China? The author should show information that it is necessary that they choose this fish skin for this study. What is the meaning of this study for the European eel or other fish aquaculture industry in China?

2.     Materials and Methods

I suggest the author add “data analysis” as part 2.5.

The author show data as “mean ± standard deviation” in Fig 5 a-c. However, there is no “data analysis” information in this part. I think this is not reasonable. The author did not conduct a significant difference analysis in all the Fig. If the data has no significant difference, the author should give necessary explanation. However, I cannot see any information on this part.

3.         Results

in Fig 5 a-c, The growth curve values are exhibited as the mean ± standard deviation (SD), n = 3. What about the significant difference between groups? There is no any information to explain

here. 

5. Discussion and Conclusions

This part was prepared well

6. Check the Reference again.

Comments on the Quality of English Language

The English could be improved to more clearly express the research.

Author Response

Comments 1:

1)    General concept comments.

I think the author should seek help with writing in English, I hope round of editing with English native language would further improve the quality of the writing. The English grammar should be checked again.

Response 1: Thank you for pointing this out. We agree with this comment. Therefore, we applied for a professional rapid English language editing service of MDPI Author Service before the revision was submitted.

Comments 2:

2)    Specific comments

Title “A reversible 2D-3D culture model from European eel (Anguilla anguilla) skin with susceptibility for different aquatic viruses”

“2D-3D” is not clear, I suggest it is better change “2D to “adherent”, and change “3D” to “suspension”. The author may consider better title such as “A reversible culture model between adherent and suspension from European eel (Anguilla anguilla) skin with susceptibility for different aquatic viruses”

Response 2: Agree. I have, accordingly, modified the title as suggested to emphasize this point, and these texts were also modified as follows:

Line 19, 28, 308, 506: “2D and 3D” has been modified as “adherent and suspension”

Line 87, 444: the texts “2D-3D” has been modified as “adherent-suspension”

The paragraphs containing “2D (adherent) and 3D (suspension)” has been completely rewritten.

Comments 3: 

  1. Introduction 

In “Introduction”, the author introduced mainly focused on “2D” and “3D”. However, there is little information on European eel (Anguilla anguilla). “In the present study, a micro-carrier-free culture system originated from European eel (Anguilla anguilla) skin has been introduced”

Why the author chose the skin of European eel to do this study, but not other fish?

What about the aquaculture output and advantage of European eel in China? The author should show information that it is necessary that they choose this fish skin for this study. What is the meaning of this study for the European eel or other fish aquaculture industry in China?

Response 3: This study of establishment of the ES cell line is a part of our long-term research project “Somatic cell Bank of characteristic economic fish in Fujian province”, which has been running since 2007. An interim appraisal report (2007-2021) of the progress of this project can be accessed at www.tech110.net with the Registration Number of Scientific and Technological Achievements: 9352022Y0029.

In 2006, Chinese aquaculture eel annual production accounts for about 2/3 of the world's total production, and the aquaculture eel production of Fujian province ranks the top in China. However, there were only several cell lines originated from Anguillidae species before 2006 (see reference 24), and none of them were created from A.anguilla. Therefore, when the cell bank project began at 2007, we chose A. anguilla as one of the major subjects, and the skin, as one of the usually target organ for all important viruses of Anguilla sp., was picked for cell line establishment.

The primary culture and the early subcultures of the ES cells (passage 2-20) were finished during 2007-2009, along with the culture of cells from fin, kidney, liver and muscle (unpublished) of A.anguilla. Elvers of two different sizes were used in the animal experiments, and these studies was funded by the Youth Project of Fujian Academy of Agricultural Sciences (A2007QJ06), and approved by the Institutional Animal Care and Use Ethics Committee of Fujian Academy of Agricultural Sciences under number BI-AEC-0007055002, the ethic approval has been enclosed (see attachment). The skin cells were then stored in liquid nitrogen, and examined annually for aliveness. In 2023, the cells in cryopreservation were recovered to accomplish the establishment of the ES cell line, these studies were approved under number BI-AEC-20230208, the ethic approval has been also enclosed (see attachment). Our studies about the establishment of other 4 A. anguilla cell lines and AngHV disease of A.anguilla have been cited in the manuscript (see references 24-27, 37, 51), and the works of my colleagues on Anguilla rostrata have been listed at the end of the response.

We didn’t include the project A2007QJ06 in the funding list when the manuscript was first submitted because it had been concluded in 2009, we were afraid that an interval of 15 years between this project and the others would be confusing. Since the history and progress have been mentioned in the revision, we’ve updated the “Funding” part and the “Institutional Review Board Statement” part as described, the project number and ethic approval number have been appended.

Because reviewer 3 and reviewer 4 gave their comments about the “Introduction” part also, this part has been totally rewritten incorporating all comments.   

Comments 4 

  1. Materials and Methods

I suggest the author add “data analysis” as part 2.5.

The author show data as “mean ± standard deviation” in Fig 5 a-c. However, there is no “data analysis” information in this part. I think this is not reasonable. The author did not conduct a significant difference analysis in all the Fig. If the data has no significant difference, the author should give necessary explanation. However, I cannot see any information on this part.

Response 4: Thank you for this valuable comment. A new subsection “Data analysis” numbered as 2.5 has been added according to the comments of reviewer 2, also a relevant new figure numbered as Figure 6 has been added to the section “Results”. Accordingly, the description of Figure 6 has been added to the subsection 3.2.2, and the part “Discussion” has been supplemented also. The numbers of original Figure 6 and Figure 7 have been changed to Figure 7 and Figure 8.

Comments 5: 

  1.  Results

in Fig 5 a-c, The growth curve values are exhibited as the mean ± standard deviation (SD), n = 3. What about the significant difference between groups? There is no any information to explain here. 

Response 5: Thank you for this valuable comment. A new histogram has been graphed as suggested. This new figure has been numbered as Figure 6 and supplemented to the manuscript, the numbers of original Figure 6 and Figure 7 have been changed to Figure 7 and Figure 8, respectively. The description of the analysis has been supplemented to the subsection 3.2.2 in line 365-382.

Comments 6

  1. Discussion and Conclusions

This part was prepared well

Response 6: Thank you so much for your approval.

Comments 7

  1. Check the Reference again.

Response 7: Agree. We’ve checked the Reference part, and found that some domain codes haven’t been deleted completely. That’s why most of the formatting errors were generated. We’ve made all the domain codes removed, and re-documented the References according to the MDPI templates. 

Comments 8

Comments on the Quality of English Language

The English could be improved to more clearly express the research.

Response 8: Agree. We’ve checked the manuscript through for grammar mistakes, and applied for a professional rapid language editing service of MDPI Author Service before the revision was submitted. 

Appendix:

Zhang LJ, Chen Q, Yang JX, Ge JQ. Immune responses and protective efficacy of American eel (Anguilla rostrata) immunized with a formalin-inactivated vaccine against Anguillid herpesvirus. Fish Shellfish Immunol. 2024 Jan;144:109262. doi: 10.1016/j.fsi.2023.109262.

Supplemented References:

  1. Haenen, O.; van Ginneken, V.; Engelsma, M.; van den Thillart, G. Impact of Eel Viruses on Recruitment of European Eel. In ed.; van den Thillart, G.; Dufour, S.; Rankin, J.C.,Eds.; Springer Netherlands: Dordrecht, 2009; pp. 387-400. doi: 10.1007/978-1-4020-9095-0_16
  2. Bryson, S.P.; Joyce, E.M.; Martell, D.J.; Lee, L.E.; Holt, S.E.; Kales, S.C.; Fujiki, K.; Dixon, B.; Bols, N.C. A Cell Line (HEW) from Embryos of Haddock (Melanogrammus aeglefinius) and its Capacity to Tolerate Environmental Extremes. Mar Biotechnol (NY) 2006, 8, 641-653. doi:10.1007/s10126-005-6163-1
  3. Zhao, Z.; Li, F.; Ning, J.; Peng, R.; Shang, J.; Liu, H.; Shang, M.; Bao, X.Q.; Zhang, D. Novel Compound FLZ Alleviates Rotenone-Induced PD Mouse Modelby Suppressing TLR4/MyD88/NF-kappaB Pathway through Microbiota-Gut-Brain Axis. Acta Pharm Sin B 2021, 11, 2859-2879. doi:10.1016/j.apsb.2021.03.020
  4. Merten, O.W. Introduction to Animal Cell Culture Technology-Past, Present and Future. Cytotechnology2006, 50, 1-7. doi:10.1007/s10616-006-9009-4
  5. Schmidt, J. Breeding Places and Migrations of the Eel. Nature1923, 111, 51-54. https://www.nature.com/articles/111051a0.pdf
  6. Arai, T. Do we Protect Freshwater Eels Or Do we Drive them to Extinction? Springerplus2014, 3, 534. doi:10.1186/2193-1801-3-534

Reviewer 3 Report

Comments and Suggestions for Authors

The study is innovative and compelling, but improvements in the Simple Summary and Abstract sections are necessary. The Simple Summary currently provides only literature information, while the Abstract focuses solely on methods and results. The Abstract should follow a structure including a brief literature review, methods, results, and conclusions. The Simple Summary should succinctly present the background, methods, and main results.

Additionally, I have noted some writing issues that affect readability, such as spacing errors (e.g., lines 42, 49, 51). I recommend addressing these and conducting a thorough review across the entire manuscript.

Further specific suggestions include:

  • Correcting the writing style of A. anguilla throughout the manuscript.
  • Clearly referencing the methods used for primary culture, subculture, and 18S rRNA sequence analysis.

I believe this study has significant potential and would merit publication once the recommended revisions are made.

Author Response

Comments 1: Comments and Suggestions for Authors

The study is innovative and compelling, but improvements in the Simple Summary and Abstract sections are necessary. The Simple Summary currently provides only literature information, while the Abstract focuses solely on methods and results. The Abstract should follow a structure including a brief literature review, methods, results, and conclusions. The Simple Summary should succinctly present the background, methods, and main results.

Response 1: Agree. Because reviewer 4 gave some comment about of the “Simple summary” part and the “Abstract” part also, these two parts have been rewritten incorporating all comments.

Comments 2:

Additionally, I have noted some writing issues that affect readability, such as spacing errors (e.g., lines 42, 49, 51). I recommend addressing these and conducting a thorough review across the entire manuscript.

Response 2: Agree. We’ve checked the manuscript through for grammar mistakes, and applied for a professional rapid language editing service of MDPI Author Service before the revision was submitted. 

Comments 3:

Further specific suggestions include:

  • Correcting the writing style of  anguillathroughout the manuscript.
  • Clearly referencing the methods used for primary culture, subculture, and 18S rRNA sequence analysis.

I believe this study has significant potential and would merit publication once the recommended revisions are made.

Response 3: Agree. We have corrected all texts containing “A.anguilla” with spacing errors, and the references have been cited as recommended in Line 95, 112, 114 and 214. The new references have been listed in the Appendix.

Line 95: Added “Explant outgrowth method was applied to initiate primary culture [28].” at the beginning of the paragraph.

Line 112: Added “In accordance with our previous studies [24,27],” at the beginning of the paragraph.

Line 130: Reference 31 was cited.

Line 241: Reference 38 was cited.

Appendix

Supplemented References:

  1. Haenen, O.; van Ginneken, V.; Engelsma, M.; van den Thillart, G. Impact of Eel Viruses on Recruitment of European Eel. In ed.; van den Thillart, G.; Dufour, S.; Rankin, J.C.,Eds.; Springer Netherlands: Dordrecht, 2009; pp. 387-400. doi: 10.1007/978-1-4020-9095-0_16
  2. Bryson, S.P.; Joyce, E.M.; Martell, D.J.; Lee, L.E.; Holt, S.E.; Kales, S.C.; Fujiki, K.; Dixon, B.; Bols, N.C. A Cell Line (HEW) from Embryos of Haddock (Melanogrammus aeglefinius) and its Capacity to Tolerate Environmental Extremes. Mar Biotechnol (NY) 2006, 8, 641-653. doi:10.1007/s10126-005-6163-1
  3. Zhao, Z.; Li, F.; Ning, J.; Peng, R.; Shang, J.; Liu, H.; Shang, M.; Bao, X.Q.; Zhang, D. Novel Compound FLZ Alleviates Rotenone-Induced PD Mouse Modelby Suppressing TLR4/MyD88/NF-kappaB Pathway through Microbiota-Gut-Brain Axis. Acta Pharm Sin B 2021, 11, 2859-2879. doi:10.1016/j.apsb.2021.03.020
  4. Merten, O.W. Introduction to Animal Cell Culture Technology-Past, Present and Future. Cytotechnology2006, 50, 1-7. doi:10.1007/s10616-006-9009-4
  5. Schmidt, J. Breeding Places and Migrations of the Eel. Nature1923, 111, 51-54. https://www.nature.com/articles/111051a0.pdf
  6. Arai, T. Do we Protect Freshwater Eels Or Do we Drive them to Extinction? Springerplus2014, 3, 534. doi:10.1186/2193-1801-3-534

Reviewer 4 Report

Comments and Suggestions for Authors

Riew comments
Simple Summary.
1. Page 1, line 8,‘The 2D and 3D cell culture methods have their own advantages.’ Could the
authors please briefly describe the differences between the advantages of the 2D and 3D cell
culture methods so that the reader can better understand them? the reader to have a better
understanding.
Abstract.
2. On page 1, lines 23-24,‘we've developed a reversible 2D-3D culture model originated from
Anguilla anguilla skin’ by what method? ‘reversible 2D-3D culture model originated’ Please
provide a brief description. Please add the author's comments.
Introduction
3. Page 1, lines 45-58,“Various methods have been tried...... direction for further cost reduction. ”
The logic is not strong enough, it is all stacked with literature, so the author is asked to re-frame
the writing.
4. Page 2, lines 59-60,‘a micro-carrier-free culture system originated from European eel (Anguilla
anguilla) skin’, the authors should consult recent years' relevant literature to give a general
introduction to it, instead of writing the findings of this paper.
5. Page 2, lines 69-73,‘To date, there is no aquatic cell line ...To date, there is no aquatic cell line’,
the authors are requested to re-frame the section. For example: ‘To solve the ..... problem, it was
therefore researched through the .... methodology is investigated with a view to generating .... in
the .... respect to have .... significant impact’
6. The entire ‘Introduction’ section needs to be carefully reconstructed and revised by the author.
Materials and Methods
7. On page 4, line 158, does the formula need to be explained by adding
relevant comments on ‘T, t’, etc.?
8. Page 5, lines 169-179, ‘Cell transfection’ Where does this part of the method originate from and
is it supported by the literature?
9. Page 5, lines 201-207, ‘Electron microscopy’ Where does the methodology in this section
originate and is it supported by the literature?
10. What software tools and methods were used to analyse the data, and could the authors please
add to this section?
Discussion
11. Page 12, lines 363-374, ‘In all batch culture tests .....for serum-free-culture adaptation’, the
results and speculations of this study should be discussed with reference to the relevant literature
and the results of other scholars, so as to summarise the insights of the present study, and the
authors are requested to re-conceptualise and revise the content of this section.
12.Page 13,lines 397-404,“As a centuried popular economic species,.... of cellular agriculture.” At
the end of the paragraph, there should be a summary statement related to the content of the
paragraph, which the author is requested to add.
13. The ‘Discussion’ section needs to be discussed and reflected upon in the same order as the
results section, and ultimately the study's own insights should be drawn.
Conclusions
14. Page 13, lines 422-423, ‘All these results suggested ...... or cellular agriculture.’ The
significance of this study has not been adequately expressed, so the authors should revise it.
15. In the conclusion section, the advantages of ‘an aquatic culture system’ in this study are
described, but it seems that there is no mention of them in the discussion, so the authors are
requested to consider whether it is necessary to add this section.
16. Please check the manuscript carefully for grammatical errors and logical relationships.

Author Response

Comments 1:

Simple Summary.

  1. Page 1, line 8,‘The 2D and 3D cell culture methods have their own advantages.’ Could the

authors please briefly describe the differences between the advantages of the 2D and 3D cell

culture methods so that the reader can better understand them? the reader to have a better

understanding.

Response 1: Thank you for pointing this out. According to the suggestion of reviewer 2, “adherent” and “suspension” are more precise to describe the two different growth modes of ES cells in this study. Therefore, we modified the title as suggested to emphasize this point, and these texts were also modified as follows:

Line 19, 28, 308, 506: “2D and 3D” has been modified as “adherent and suspension”

Line 87, 444: the texts “2D-3D” has been modified as “adherent-suspension”

The paragraphs containing “2D (adherent) and 3D (suspension)” has been completely rewritten.Because reviewer 3 gave some comments about the “Simple Summary” part also, this part has been totally rewritten incorporating all comments.

Comments 2: 

Abstract.

  1. On page 1, lines 23-24,‘we've developed a reversible 2D-3D culture model originated from

Anguilla anguilla skin’ by what method? ‘reversible 2D-3D culture model originated’ Please

provide a brief description. Please add the author's comments.

Response 2: Thank you for this valuable comment. Because reviewer 3 gave some comments about the “Abstract” part also, this part has been rewritten incorporating all comments.

Comments 3: 

Introduction

  1. Page 1, lines 45-58,“Various methods have been tried...... direction for further cost reduction. ”

The logic is not strong enough, it is all stacked with literature, so the author is asked to re-frame

the writing.

Response 3: Agree. Because reviewer 2 gave some comments about the “Introduction” part also, this part has been utterly rewritten incorporating all comments.

Comments 4: 

  1. Page 2, lines 59-60,‘a micro-carrier-free culture system originated from European eel (Anguilla

anguilla) skin’, the authors should consult recent years' relevant literature to give a general

introduction to it, instead of writing the findings of this paper.

Response 4: Thank you for this valuable comment. The whole paragraph has been rewritten as suggested.

Comments 5: 

  1. Page 2, lines 69-73,‘To date, there is no aquatic cell line ...To date, there is no aquatic cell line’,

the authors are requested to re-frame the section. For example: ‘To solve the ..... problem, it was

therefore researched through the .... methodology is investigated with a view to generating .... in

the .... respect to have .... significant impact’.

Response 5: Thank you for this comment. This paragraph has been rewritten as below:

Line 82-91: “In the present study, to build an efficient in vitro culture model for virus isolation and vaccine development of Anguilla sp., a new A. anguilla skin cell line ES has been established and characterized. A long-term time course growth observation has been conducted using batch culture and simulated semi-continuous culture processes to determine the suspension features of the ES cell line, presenting its advantages as a natural reversible adherent-suspension culture model. The infection tests confirmed its susceptibility to both AngHV and Red-spotted Grouper Nervous Necrosis Virus (RGNNV), two important aquatic viruses in Chinese aquaculture industry. The ES cell line is supposed to be potential for production of aquatic vaccines, and also other cell-based protein products.”

Comments 6: 

  1. The entire ‘Introduction’section needs to be carefully reconstructed and revised by the author.

Response 6: Thank you for this valuable comment. The section “Introduction” has been completely rewritten as recommended.

Comments 7: 

Materials and Methods

  1. On page 4, line 158,

does the formula need to be explained by adding

relevant comments on ‘T, t’, etc.?

Response 7: Agree. Comments has been added to explain the formula as follows:

Line 178-181: “…(T means the population doubling time of the logarithmic phase, Nt represents the final number of cells, N0 represents the initial number of cells, and t represents the time interval between Nt and N0)…”

Comments 8: 

  1. Page 5, lines 169-179, ‘Cell transfection’ Where does this part of the method originate from and

is it supported by the literature?

Response 8: Thank you for this comment. This part of method originate from the “lipofectamine™ 3000 reagent protocol” (Invitrogen) with some modifications.

Line 201-202: the text “according to the official protocol with some modifications, ” has been supplemented.

Comments 9: 

  1. Page 5, lines 201-207, ‘Electron microscopy’ Where does the methodology in this section

originate and is it supported by the literature?

Response 9: Thank you for this comment. We have contacted with Servicebio Technology Co., and supplemented a new reference as ref. 38. It’s been cited in line 241.

Comments 10: 

  1. What software tools and methods were used to analyse the data, and could the authors please

add to this section?

Response 10: Thank you for pointing this out. A new subsection “Data analysis” numbered as 2.5 has been added according to the comments of reviewer 2, also a relevant new figure numbered as Figure 6 has been added to the section “Results”. Accordingly, the description of Figure 6 has been added to the subsection 3.2.2, and the part “Discussion” has been supplemented also. The numbers of original Figure 6 and Figure 7 have been changed to Figure 7 and Figure 8.

Comments 11:  

Discussion

  1. Page 12, lines 363-374, ‘In all batch culture tests .....for serum-free-culture adaptation’, the

results and speculations of this study should be discussed with reference to the relevant literature

and the results of other scholars, so as to summarise the insights of the present study, and the

authors are requested to re-conceptualise and revise the content of this section.

Response 11:Thank you for pointing this out. The section “Discussion” has been completely rewritten as recommended.

Comments 12:  

12.Page 13,lines 397-404,“As a centuried popular economic species,.... of cellular agriculture.” At

the end of the paragraph, there should be a summary statement related to the content of the

paragraph, which the author is requested to add.

Response 12: Agree. We have specifically emphasized this point in the rewrite.

Comments 13: 

  1. The ‘Discussion’ section needs to be discussed and reflected upon in the same order as the

results section, and ultimately the study's own insights should be drawn.

Response 13: Thank you for this valuable comment. We agree with this comment, and the part “Discussion” has been completely rewritten as recommended.

Comments 14: 

Conclusions

  1. Page 13, lines 422-423, ‘All these results suggested ...... or cellular agriculture.’ The

significance of this study has not been adequately expressed, so the authors should revise it.

Response 14: Thank you for this valuable comment. We’ve emphasized in the conclusion that ES is the first aquatic-origin natural suspension culture system, and rewritten the last sentence as:

Line 509-512:“…All these results suggest that the ES culture system is more cost-effective by removing the processes of suspension domestication and carrier dissolution, which can benefit the production of aquatic vaccines, biopharmaceutics, or cellular agriculture.”

Comments 15: 

  1. In the conclusion section, the advantages of ‘an aquatic culture system’ in this study are

described, but it seems that there is no mention of them in the discussion, so the authors are

requested to consider whether it is necessary to add this section.

Response 15: Agree. We’ve check the manuscript through and moved the description of the trends and advantages of an aquatic culture system from the part “introduction” to the part “Discussion” accordingly.

Comments 16: 

  1. Please check the manuscript carefully for grammatical errors and logical relationships.

Response 16: Agree. We’ve checked the manuscript through for grammar mistakes, and applied for a professional rapid language editing service of MDPI Author Service before the revision was submitted. 

Appendix

Supplemented References:

  1. Haenen, O.; van Ginneken, V.; Engelsma, M.; van den Thillart, G. Impact of Eel Viruses on Recruitment of European Eel. In ed.; van den Thillart, G.; Dufour, S.; Rankin, J.C.,Eds.; Springer Netherlands: Dordrecht, 2009; pp. 387-400. doi: 10.1007/978-1-4020-9095-0_16
  2. Bryson, S.P.; Joyce, E.M.; Martell, D.J.; Lee, L.E.; Holt, S.E.; Kales, S.C.; Fujiki, K.; Dixon, B.; Bols, N.C. A Cell Line (HEW) from Embryos of Haddock (Melanogrammus aeglefinius) and its Capacity to Tolerate Environmental Extremes. Mar Biotechnol (NY) 2006, 8, 641-653. doi:10.1007/s10126-005-6163-1
  3. Zhao, Z.; Li, F.; Ning, J.; Peng, R.; Shang, J.; Liu, H.; Shang, M.; Bao, X.Q.; Zhang, D. Novel Compound FLZ Alleviates Rotenone-Induced PD Mouse Modelby Suppressing TLR4/MyD88/NF-kappaB Pathway through Microbiota-Gut-Brain Axis. Acta Pharm Sin B 2021, 11, 2859-2879. doi:10.1016/j.apsb.2021.03.020
  4. Merten, O.W. Introduction to Animal Cell Culture Technology-Past, Present and Future. Cytotechnology2006, 50, 1-7. doi:10.1007/s10616-006-9009-4
  5. Schmidt, J. Breeding Places and Migrations of the Eel. Nature1923, 111, 51-54. https://www.nature.com/articles/111051a0.pdf
  6. Arai, T. Do we Protect Freshwater Eels Or Do we Drive them to Extinction? Springerplus2014, 3, 534. doi:10.1186/2193-1801-3-534

Round 2

Reviewer 2 Report

Comments and Suggestions for Authors

The author has revised the paper according to the point-to-point requirements from review. 

Comments on the Quality of English Language

The English could be improved to more clearly express the research